# Specificity of the Associations between Indices of Cardiovascular Health with Health Literacy and Physical Literacy; A Cross-Sectional Study in Older Adolescents

**DOI:** 10.3390/medicina58101316

**Published:** 2022-09-20

**Authors:** Marijana Geets Kesic, Anamarija Jurcev Savicevic, Mia Peric, Barbara Gilic, Natasa Zenic

**Affiliations:** 1Faculty of Kinesiology, University of Split, 21000 Split, Croatia; 2Teaching Institute of Public Health of Split Dalmatian County, 21000 Split, Croatia; 3Department of Health Studies, University of Split, 21000 Split, Croatia; 4School of Medicine, University of Split, 21000 Split, Croatia; 5Faculty of Kinesiology, University of Zagreb, 10000 Zagreb, Croatia

**Keywords:** health behaviors, lipid profile, public health, youth

## Abstract

*Background and Objectives*: Cardiovascular health status (CVHS) is an important determinant of health, while it is theorized that health literacy (HL) and physical literacy (PL) could be directly related to CVHS. The aim of this study was to evaluate gender-specific associations between PL and HL and indices of CVHS in adolescence. *Materials and Methods*: The participants were 247 adolescents (177 females) from Split-Dalmatia county in Croatia who were tested on HL, PL, and CVHS (physical activity level (PAL) and lipid profile). The lipid profile included total cholesterol, triglycerides, high-density lipoproteins, non-high-density lipoprotein-cholesterol, and low-density lipoproteins. Gender-stratified multivariate cluster analysis (K-means clustering) was used to group participants into three homogenous groups on the basis of their HL and PL, while differences between clusters in CVHS were evidenced by analysis of the variance and consecutive post-hoc tests. *Results*: The lipid profile was better in girls with higher HL scores. Additionally, clusters consisting of participants with a better PL were characterized by higher PAL. We have found no evidence that HL is associated with PAL, while PL was not associated with the lipid profile. *Conclusions*: HL was specifically associated with direct indicators of health status (lipid profile) in girls, while PL was associated with PAL as a particular behavioral health indicator in both genders. The study highlights the necessity of including education of HL and PL in schools.

## 1. Introduction

Health literacy (HL) and physical literacy (PL) are concepts that are associated with health behaviors and outcomes [1,2], which is especially important during adolescence, as adolescents form their health habits during this life period [3]. HL entails “people’s knowledge, motivation, and competencies to access, understand, appraise and apply health information to make judgments and take decisions in everyday life concerning healthcare, disease prevention and health promotion to maintain or improve quality of life during the life course” [4]. Numerous studies report that HL directly influences the health of adolescents. For example, high school students with higher HL had better health-promoting behaviors (i.e., good nutritional habits, not using psychoactive substances, higher physical activity), health outcomes, and better health-related quality of life [5,6,7,8]. On the other side, PL, defined as “the motivation, confidence, physical competence, knowledge, and understanding to value and take responsibility for engagement in physical activities for life” [9], is also connected to positive health outcomes, mainly through fostering involvement in physical activities [1]. Briefly, PL has been associated with various health indicators, including cardiorespiratory fitness, physical activity, and health-related quality of life [1,10]. However, a recent study on Croatian adolescents reported that HL and PL, although both related to health behaviors and outcomes, are not interrelated and should be assessed separately in evaluating health behaviors [11].

Having adequate and maintaining physical activity level (PAL) is an essential factor that influences health status, including cardiovascular health status (CVHS) [12,13]. As engaging in physically demanding activities is one of the most important positive health behaviors, high levels of both HL and PL are essential for reaching adequate PAL [10,14]. However, the majority of adolescents worldwide (81%) do not have a sufficient PAL [15], which deteriorates their present and future health, including CVHS [16]. Moreover, adolescents were recently exposed to movement restrictions as a preventive measure for the COVID-19 pandemic, which decreased their PAL even more. Indeed, numerous studies around the globe reported a decrease in PAL in adolescents as a result of the COVID-19 lockdown [17,18,19,20]. Consequently, a decline in PAL due to the COVID-19 pandemic deteriorated adolescents’ health [21]. Thus, concepts related to improving skills necessary for maintaining health habits (i.e., HL and PL) became even more important [22].

The lipid profile is one of the most important determinants of CVHS [23,24]. In most common words, lipid profile describes levels of lipids in the blood and most commonly consists of high-density lipoprotein (HDL) cholesterol, low-density lipoprotein (LDL) cholesterol, and triglycerides [25]. High blood LDL cholesterol levels and triglycerides indicate an increased risk of cardiovascular disease, while increased HDL cholesterol levels indicate a healthy cardiovascular system [26]. Physical activity and exercise improve lipid profiles by enhancing the ability of muscles to use lipids as opposed to glycogen, leading to reduced plasma lipid levels [27]. Supportively, several studies reported that children and adolescents with higher PAL had better lipid profiles than those with low PAL. Precisely, a study on youth aged 9–18 years recorded that PAL and physical fitness are strongly associated with the lipid profile and a lower risk for coronary heart disease [28]. Moreover, children and adolescents aged 6–17 years who reached recommended daily PAL had more favorable HDL cholesterol [29]. Thus, as PAL is known to be linked to both HL and PL, it is expected that adolescents with better PL and HL will also have a better/healthier lipid profile. However, studies directly investigating the associations between HL and PL and lipid profiles in adolescents are missing.

From the previous literature overview, it is clear that better HL and PL should be linked to positive health behaviors even in adolescence, and it is expected that adolescents with better HL and PL will have better PAL and lipid profiles. However, studies done so far have rarely examined all indices simultaneously, and there is a lack of comprehensive knowledge regarding how HL and PL correspond to PAL and lipid profile in adolescents. Thus, the main aim of this study was to investigate the associations between HL and PL (on one side) and the lipid profile and PAL (on the other side). Since previous studies reported significant differences between males and females in study variables [15,30,31,32,33], we decided to make a more methodologically correct and precise investigation and to determine the gender-stratified associations between HL and PL and indices of CVHS (PAL and lipid profile). We hypothesized that adolescents with better HL and PL would generally have better CVHS, with some gender differences in established associations.

## 2. Materials and Methods

### 2.1. Participants and Study Design

The study had a cross-sectional character. The study included high school students (in total 247 participants), of which 70 were male, 177 were female, and average age was 16.8 ± 1.3 years. The location for the study was a training college for medical students in Split, Croatia. A minimum sample size of 85 participants was established based on a correlation between HL and PL of 0.30 suggested in a pilot study on Croatian college students, with a type-I error rate of 0.05 and a type-II error rate of 0.20 [34].

The study permitted the inclusion of participants with their mean age falling within the World Health Organization’s (WHO) definition of adolescents (10–19 years of age). Potential participants with (or suspected of) acute inflammatory disease (e.g., COVID-19) at the time of testing were excluded from the study. The Ethical Board of the University of Split, Faculty of Kinesiology, approved the study on 23 September 2021 (EBO: 2181-205-02-01-21-0011).

Successive to ethical approval, the scope and procedure for the study were presented to all classes targeted to participate by one of the first authors. Students (or parents/legal guardians for those younger than 18 years of age) were requested to sign a written consent form to participate before starting the study. In total, more than 401 consent waivers were distributed, of which 289 were recovered, with a response rate of 72%. In this study, we included participants who were successfully tested for all variables observed (please see later for details). Participants were informed that they had the right to withdraw from the study anytime. Measuring was conducted during May and April 2022 during school hours (08:00 a.m. to 10:00 a.m.).

### 2.2. Variables and Measurement

HL, PL, PAL, standard anthropometric measures, and lipid profiles were included as variables in this study.

The basis for evaluating HL was the European Health Literacy Survey Questionnaire (HLS-EU-Q), developed by Sørensen, et al. [35]. The questionnaire comprises 47 questions measuring an individual’s capacity to obtain, process, and understand basic health information and services to make appropriate health decisions or to access, understand, appraise, and apply health-related information. A 4-point Likert scale, with responses from very difficult—1 to very easy—4, was used to construct a general index of HL. The formula: index = (mean − 1) × (50/3) was employed to calculate the score. An HL scale of 0–50 was created, considering 0 as the lowest score and 50 as the highest. The index was split into four bands of HL as follows: inadequate (from 0 to 25); problematic (26–33); sufficient (34–42); excellent (43–50). In this study, we used the Croatian version of the HLS-EU-Q47 questionnaire, which was previously shown to be reliable and valid among Croatian adolescents) [34].

To evaluate the current level of PL, the PLAYself questionnaire was used. This is a self-assessment tool and considers four main groups of questions: (i) the affective and cognitive aspect of PL; (ii) the environmental ability; (iii) the estimation of literacy, numeracy, and physical literacy in different settings; (iv) fitness. Final scoring for the assessment is made by combining the sum of the first three groups to obtain a total and then dividing by the number of questions asked [36]. A total score of 100 indicated the maximum self-perceived PL. The study employed the Croatian version of the PLAYself questionnaire, which produced reliable results in previous works regarding Croatian adolescents [37,38].

PAL was evaluated using the Physical Activity Questionnaire for Youth (PAQ-A). The PAQ-A consists of 9 questions regarding physical activity habits, answered on a 5-point Likert scale (1-no/low activity, 5-high activity) [39].

The platform SurveyMonkey (SurveyMonkey Inc., San Mateo, CA, USA) was used to carry out all questionnaires, HLS-EU-Q47, PLAYself, and PAQ-A.

The anthropometric measures included body mass (to the nearest 0.1 kg) and body height (to the nearest cm). The anthropometric measurement took place in the school laboratory with strict privacy conditions every morning from 08:00 to 10:00. The procedure and protocols were explained to each student preceding the measurement. During the measurement, students were dressed in their underwear and barefoot.

The participant’s/student’s lipid profiles consisted of total cholesterol (TCHOL), triglycerides (TG), high-density lipoprotein (HDL-C), non-high-density lipoprotein-cholesterol (non-HDL-C), low-density lipoproteins (LDL-C), and the CHOL/HDL ratio. To evaluate this study’s lipid profile, we used point-of-care testing (POCT). POCT is a minimally invasive diagnostic method that has the potential to provide rapid and accurate results [40]. The testing is based on reactive strips consisting of a membrane that removes the red blood cells, while plasma lipids are determined by a dry chemical reaction. In this study, we used Mission Cholesterol Test Devices (ACON Laboratories, Inc., San Diego, CA, USA). The device uses total capillary blood samples. The 3-in-1 Lipid Panel Strips from the same producer simultaneously measures the concentration of total cholesterol (TCHOL), high-density lipoprotein (HDL), and triglycerides (TRIG), while the LDL and CHOL/HDL ratio are automatically calculated. Several recent studies recommended calculating non-HDL-C because it has more atherogenic properties than all other lipoproteins [41,42]. Sigdel, et al. [43] found in a multinational study that a high concentration of non-HDL-C can predict cardiovascular risk [43]. Non-HDL-C is calculated as the total cholesterol minus HDL-C [44].

The guidelines of the manufacturer’s protocol were observed. In brief, before starting sampling, each new box of the test device was calibrated by inserting a cod chip that automatically calibrates the meter. Further, all students were instructed to undergo an 8–12 h fasting period before the examination. The finger puncture to collect a capillary blood drop was performed in the School Hematology Laboratory by laboratory technicians supervised by a medical doctor. The optimal operating temperature was set at 20 °C. The finger punction each student was informed by a medical doctor about the procedures, risks, benefits, and rights, and all their doubts were clarified. To transfer the fresh capillary blood sample to the test strips in the right volume (35 µL), we used a capillary transfer tube (ACON, Mission, San Diego, CA, USA).

According to the National Cholesterol Education Program (NCEP) Expert Panel on Cholesterol Levels in Children and Adolescents [45], cut-off values for plasma lipid and lipoprotein levels were used, as presented in Table 1.

### 2.3. Statistical Analyses

All variables were checked for normality of the distributions by the Kolmogorov–Smirnov test, and means and standard deviations were reported. Plasma lipid and lipoprotein levels are reported in percentages according to cut-off values (please see Table 1 for details).

In the first phase, we calculated Spearman’s correlation coefficients between all observed variables. Next, a multivariate cluster analysis (K-means clustering method), with a predefined number of clusters (three) was performed to identify the member of three homogenous groups of participants on the basis of their HL and PL (i.e., the multivariate design allows simultaneous observation of the results on both applied variables). Although this statistical procedure performs grouping into two clusters by default, the number of three clusters was predefined to avoid simplified grouping into two clusters (high PL and HL vs. low HL and PL) and to allow an eventual, more complex identification of participants’ characteristics (see later result for more details). In the next phase, participants were allocated to clusters, and cluster characteristics with regard to grouping variables (HL and PL) were identified by a one-way analysis of variance (ANOVA).

ANOVA was used to identify the differences among clusters for each gender. Specifically, in ANOVA calculations, the allocation to each cluster was used as a “grouping variable” (categorical factor), while CVHS indices were observed as dependent variables. When ANOVA reached a statistical significance of *p* < 0.05, a Scheffe post-hoc analysis was calculated to identify the significance of the between-cluster differences.

All analyses were gender-stratified. Statistica ver. 13.5 (Tibco Inc., Palo Alto, Ca, USA) was used, and *p* < 0.05 was applied.

## 3. Results

Plasma lipid and lipoprotein status in adolescents are presented in Table 2.

Descriptive statistics for the study variables are presented in Table 3.

Apart from some logical and expected correlations between variables that were derived on the basis of calculations (i.e., correlation between non-HDL-C and LDL-C, LDL-C and CHOL/HDL ratio) in boys, some interesting associations are as follows. The TG was correlated with body mass (less than 5% of the common variance), while PL and PAL were significantly correlated in boys (17% of the common variance). Lipid panel indicators were significantly intercorrelated. Positive correlations were evidenced between TCHOL and TG (25% of the common variance), non-HDL-C (75% of the common variance), LDL-C (70% of the common variance), and CHOL/HDL ratio (51% of the common variance). TG was correlated with non-HDL-C (35% of the common variance), LDL-C (11% of the common variance), and CHOL/HDL ratio (13% of the common variance) (Table 4).

In girls, body mass was negatively correlated to HDL-C (4% of the common variance) and positively correlated to non-HDL-C (>4% of the common variance), and LDL-C (3% of the common variance). PL was significantly correlated to PAL (25% of the common variance) and HL (9% of the common variance). HL was found to be positively correlated with TCHOL (4% of the common variance), negatively correlated to LDL-C (6% of the common variance), and negatively correlated with non-HDL-C (6% of the common variance). TCHOL was positively correlated with TG (8% of the common variance), HDL-C (6% of the common variance), non-HDL-C (7% of the common variance), LDL-C (70% of the common variance), and CHOL/HDL ratio (10% of the common variance). TG was correlated with non-HDL-C (12% of the common variance) and LDL-C (3% of the common variance) (Table 5).

Cluster analysis calculated for boys based on their results achieved at HL and PL formed three characteristic homogenous groups. Cluster 1 consisted of participants who achieved low results on HL and low results on PL (low PL—low HL; L-PL/L-HL). Participants who achieved high results on HL and average results on PL were grouped into Cluster 2 (average PL—high HL; A-PL/H-HL)). Cluster 3 was formed by boys who achieved average results on HL but high results on PL (H-PL/A-HL) (Figure 1).

For boys, ANOVA identified only one significant difference between clusters in indicators of CVHS, where the L-PL/L-HL group achieved the lowest results in PAL (F-test = 17.11, *p* < 0.001; significant post-hoc difference when compared to H-PL/A-HL group (Cluster 3).

When cluster analysis was calculated for girls, Cluster 1 consisted of girls who achieved average results on PL and low results on HL (L-PL/L-HL). Girls who achieved high results on HL and high results on PL were grouped into Cluster 2 (H-PL/H-HL), while the Cluster 3 consisted of girls who achieved average results on HL and low results on PL (L-PL/A-HL) (Figure 2).

The ANOVA among clusters identified significant effects for PAL and non-HDL-C in girls (F-test = 13.11, and 11.12, respectively, both *p* < 0.001). Significant post-hoc differences for PAL were identified between Cluster 2 (H-HL/H-PL group) and Cluster 3 (L-PL/A-HL), with a better result on PAL for Cluster 1, indicating an association between higher PL with higher PAL in girls. Further, non-HDL-C was lowest in Cluster 1 (L-PL/L-HL group), with significant post-hoc differences when compared to Cluster 2 (H-HL/H-PL group), indicating an association between better HL and a more favorable lipid profile among girls.

## 4. Discussion

The aim of this study was to determine the gender-stratified associations between HL and PL and indices of CVHS (PAL and lipid profile). The study evidenced several most important findings: (i) HL was higher among girls with better lipid profile; (ii) PL was higher among adolescents with a higher PAL; (iii) HL was not associated with PAL, while PL was not associated with the lipid profile. Therefore, our initial study hypothesis was confirmed.

### 4.1. Health Literacy, Physical Literacy, and Lipid Profile

Results that adolescents with higher HL have a more favorable lipid profile are in accordance with several previous studies. Research on nursing students evidenced a positive association between HL and HDL cholesterol and a negative association of HL with cholesterol ratio (i.e., total cholesterol/HDL cholesterol), indicating a better lipid profile in students with higher HL [46]. A study on adolescents from Taiwan indirectly confirmed such associations, as it evidenced that adolescents with high HL were less likely to be obese and have poor nutritional habits (high sugar-sweetened beverage, salt, and fat intake), which are one of the main factors of having a less favorable lipid profile [47]. Meanwhile, although a recent study on Croatian adolescents did not evidence an association between body composition (body fat percentage, muscle mass) and HL [11], the authors of that study concluded that some other and more precise health indicators (i.e., lipid profile) could be more associated with HL, which was actually confirmed here. The explanation for the association between HL and lipid profile could lie in the theory that individuals with better HL scores are able and willing to take actions that improve their health, leading to a better lipid profile [4]. Another possible explanation of this association could be that individuals with better HL skills can understand the results of their blood tests, including cholesterol screening [48]. Consequently, they are able to, with the guidance of health professionals, make appropriate changes in weight, diet, and exercise to control their lipid profile.

In our study, PL was not associated with the lipid profile, which is somewhat surprising if we consider previous studies that indicated a positive association between PL and health status indices [10]. For example, a study on Canadian children evidenced a positive relationship between PL and health indicators, including body fat percentage, aerobic fitness, blood pressure, and health-related quality of life [49]. Also, another study on Canadian children evidenced better cardiorespiratory fitness in children with higher PL scores [50]. However, even though previous studies evidenced associations between PL and health indicators, none specifically investigated lipid profiles. This altogether could point out that some other aspects related to the lipid profile are more important than PL, at least in adolescence. Namely, PL is mainly related to an increasing PAL (please see the following paragraph for a more detailed explanation) and probably has limited influence on other health-related behaviors (i.e., nutritional habits, substance use) that are known to be more influential on the lipid profile [51]. Thus, as more factors could have influenced the lipid profile, maybe PL alone cannot give a precise picture of the most important determinants of the lipid profile. No association between PAL and the lipid profile in our sample (please see results for more details) could be explained by numerous factors that influenced the lipid profile, and not only PAL. It has to be mentioned that the association between HL and the lipid profile differed by gender; HL was higher among girls with a better lipid profile. This can be explained by results from a recent study on a similar sample, where it was evidenced that PL and HL were associated only among girls [11]. Therefore, it could be theorized that HL has a greater influence on health status (i.e., lipid profile) among girls, as it is also associated with another health-promoting concept—PL.

### 4.2. Physical Literacy, Health Literacy, and Physical Activity Levels

A positive association between PL and PAL was expected, as PL is generally theorized to be the foundation of participation in physical activities [52]. Indeed, studies frequently reported correlations between PAL and PL in children and adolescents. Canadian children aged 8–12 that were meeting physical activity guidelines (i.e., had sufficient PAL) had higher PL scores (physical competence, motivation, and confidence domains) than children with low PAL [53]. Another Canadian study on children aged 7–14 noted a positive association between PL (movement competence domain) and objectively measured PAL [54]. A study on Chinese adolescents aged 12–18 years also recorded a positive relationship between perceived PL and PAL [55]. What is more, a recent review study reported that PL interventions led to increased PAL, supporting the association between PL and PAL [56]. Our results also confirmed that adolescents with higher PAL possess better PL, which can be explained by the following. First, physically literate individuals are deemed to possess movement competence, confidence and motivation, knowledge and understanding of the importance of engaging in physical activities [57]. Second, physically literate individuals are willing to participate (i.e., are action-oriented) in various movements, leading to lifelong participation in physical activities and overall increased PAL [58]. Putting it all together, physically literate individuals are more likely to be engaged in various physically demanding activities, which logically increases their PAL.

On the other side, HL was not associated with PAL, which contradicts some previous studies. Namely, a study on Finnish adolescents aged 13–15 reported that adolescents with higher HL levels were more likely to participate in sports activities, leading to increased PAL [59]. Moreover, a study on 8–11 year-old children from the Netherlands recorded a strong positive relationship between HL and PAL; children with higher HL had higher PAL [60]. Meanwhile, we did not record a correlation between PAL and HL. The first possible reason for this can be found in the age of the participants. Namely, previous studies where authors evidenced a correlation between HL and PAL examined younger subjects than we did herein (8–15 and 16–18 years of age, respectively) [59,60]. Meanwhile, it is well documented that PAL decreases after the age of 14 years, mostly due to drop-outs from sports [61,62]. Therefore, it is possible that these relationships, which are evidenced in childhood and younger adolescence (i.e., higher PAL in those with better HL), are not characteristic for older adolescence simply because the HL logically increases (as a result of schooling and education), while PAL unfortunately decreases (as a result of various factors, mostly quitting sports). The indirect support for such an explanation could be found in the fact that HL and PL are actually independent qualities, at least in late adolescence [11]. Namely, even though both PL and HL are related to health behaviors (with engaging in physical activities being one of the most important ones), they are distinct concepts and evidently relate to different health behaviors. Precisely, physical activity is at the center of the PL concept, as PL is theorized to be the foundation of physical activity participation; hence it is logical that PL will have strong associations with PAL. On the other side, HL is the ability to make reasonable decisions regarding healthcare, disease prevention, and health promotion that positively influence health [4]. Therefore, it is evident that HL covers a larger area of health-related behaviors, altogether resulting in a lack of association between HL and PAL in our study.

### 4.3. Limitations and Strengths

The cross-sectional nature of the investigation is the main limitation of this study. Therefore, the cause–effect relationship between variables cannot be speculated. Moreover, HL, PL, and PAL were assessed via questionnaires, which could lead to collecting not completely honest answers. However, we tried to reduce this problem by anonymous testing, while self-selected codes were used for matching questionnaires with other variables. Moreover, another limitation can be the usage of the Mission Cholesterol Test Device, which might have analytical biases for some of the analyzed variables. Finally, lipid results were not confirmed using laboratory or additional analytical equipment, but this was a consequence of the testing protocol, which included testing in school settings and during school time. Namely, the study was done during COVID-19 pandemic, so we tried to avoid the possibility of participants being exposed to COVID-19 or any other disease in hospital/referent laboratory settings.

The main strength of this study is that it is one of the first studies that investigated HL and PL in relation to CVHS indices (lipid profile, PAL) in adolescents. Additionally, knowing that the studied indices vary across different geographical regions and cultures, it is important to note that this is probably the first study that has investigated this problem in southeastern Europe. This altogether makes the study of certain importance, as it can guide scientists and public health authorities to investigate and promote HL and PL, which would hopefully lead to improved health of adolescents.

## 5. Conclusions

This study evidenced that HL was higher among adolescents with a better lipid profiles, while HL was not correlated with PAL. On the other hand, PL was higher among adolescents with a higher PAL, while PL was not associated with lipid profiles. Thus, it could be concluded that HL is more associated with direct health indicators, while PL is more related to PAL. Due to the cross-sectional nature of the study, the causality between variables cannot be speculated, and further longitudinal studies should evaluate the true nature of evidenced relationships. In brief, while higher PAL can be a consequence of better PL (i.e., participants who are better informed on physical activities (and have better PL) will be more physically active), the opposite direction is also possible (i.e., participants who spend more time in physically demanding activities will be more physically competent and will have better PL).

This study confirms that correlates of HL and PL are relatively independent and again confirmed that HL and PL cover distinct health-related behaviors. As the issue investigated in this study is of great importance for promoting health behaviors and health in general among adolescents, it can encourage scientists to more deeply explore such concepts and include public health authorities to implement them in health promotion strategies. Also, as this study included adolescents in the education system, this study points out the necessity of including HL and PL education in schools.

## Figures and Tables

**Figure 1 medicina-58-01316-f001:**
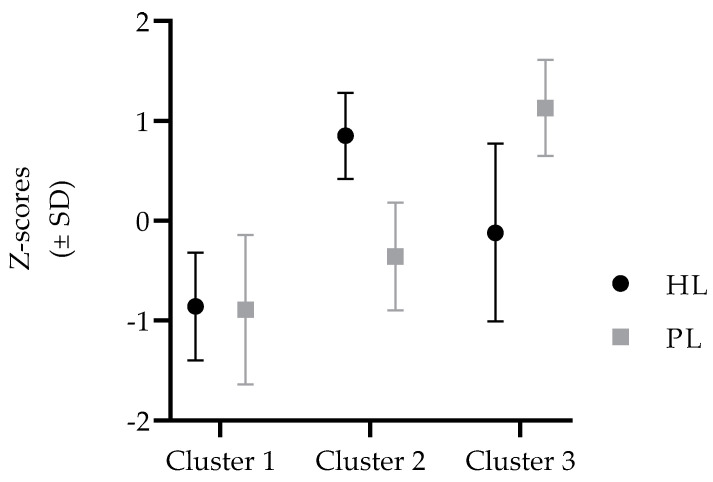
Multivariate clustering on the basis of health literacy (HL) and physical literacy (PL) for boys (results are presented in standardized values).

**Figure 2 medicina-58-01316-f002:**
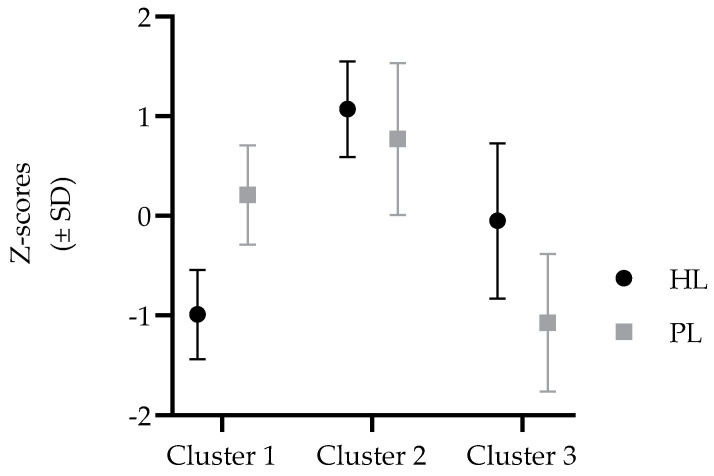
Multivariate clustering on the basis of health literacy (HL) and physical literacy (PL) for girls (results are presented in standardized values).

**Table 1 medicina-58-01316-t001:** Plasma lipid and lipoprotein level cut-off values.

Category	Acceptable mg/dL (mmol/L)	Borderline mg/dL (mmol/L)	High mg/dL (mmol/L)
TCHOL	<170 (4.4)	170 to 199 (4.4 to 5.2)	≥200 (5.2)
LDL-C	<110 (2.8)	110 to 129 (2.8 to 3.3)	≥130 (3.4)
Non-HDL-C	<120 (3.1)	120 to 144 (3.1 to 3.7)	≥145 (3.8)
TG	<90 (1 mmol/L)	90 to 129 (1 to 1.5)	≥130 (1.5)
HDL-C	<40 (1)	40 to 45 (1 to 1.2)	>45 (1.2)

Legend: TCHOL—total cholesterol, TG—triglycerides, HDL-C—high-density lipoprotein, non-HDL-C—non-high-density lipoprotein-cholesterol, LDL-C—low-density lipoproteins.

**Table 2 medicina-58-01316-t002:** Plasma lipid and lipoprotein status in studied adolescents.

	High F (%)	Borderline F (%)	Acceptable F (%)
TCHOL	2 (1)	15 (6)	230 (93)
TG	16 (7)	48 (20)	183 (73)
HDL-C	47 (19)	20 (8)	181 (73)
non-HDL-C	4 (2)	13 (5)	231 (93)
LDL-C	2 (1)	9 (4)	236 (95)

Legend: TCHOL—total cholesterol, TG—triglycerides, HDL-C—high-density lipoprotein, non-HDL-C—non-high-density lipoprotein-cholesterol, LDL-C—low-density lipoproteins.

**Table 3 medicina-58-01316-t003:** Descriptive statistics for studied variables (data are given as means ± standard deviations).

	Total Sample (*n* = 247)	Boys (*n* = 70)	Girls (*n* = 177)
Body height (cm)	1.71 ± 0.18	1.81 ± 0.14	1.68 ± 0.18
Body mass (kg)	67.01 ± 12.27	74.4 ± 12.49	64.48 ± 11.15
PAL (score)	2.51 ± 0.72	2.64 ± 0.72	2.47 ± 0.72
PL (score)	69.1 ± 11.11	69.05 ± 11.04	68.91 ± 11.40
HL (score)	37.91 ± 6.31	37.78 ± 6.53	38.07 ± 6.42
TCHOL (mg/dL)	132.58 ± 24.63	120.9 ± 23.82	136.25 ± 23.78
TG (mg/dL)	82.89 ± 29.86	82.25 ± 33.99	83.09 ± 28.53
HDL-C (mg/dL)	51.46 ± 13.68	40.12 ± 13.17	55.02 ± 11.79
non-HDL-C (mg/dL)	78.32 ± 28.02	76.94 ± 30.29	78.76 ± 27.32
LDL-C (mg/dL)	58.98 ± 30.32	52.33 ± 35.58	61.07 ± 28.25
CHOL/HDL ratio (ratio)	2.5 ± 1.6	2.48 ± 1.68	2.51 ± 1.58

Legend: PAL—physical activity level, PL—physical literacy, HL—health literacy, TCHOL—total cholesterol, TG—triglycerides, HDL-C—high-density lipoprotein, non-HDL-C—non-high-density lipoprotein-cholesterol, LDL-C—low-density lipoproteins.

**Table 4 medicina-58-01316-t004:** Correlations between studied variables for boys (*n* = 70).

	1	2	3	4	5	6	7	8	9	10
Body height (1)	−									
Body mass (2)	0.00	−								
PAL (3)	−0.10	0.02	−							
PL (4)	0.09	0.01	0.43 *	−						
HL (5)	−0.11	0.03	0.04	0.07	−					
TCHOL (6)	−0.13	0.08	0.03	−0.07	−0.03	−				
TG (7)	0.06	0.29 *	0.01	0.03	−0.06	0.53 *	−			
HDL-C (8)	0.04	−0.23	0.11	0.23	0.23	0.14	−0.20	−		
non-HDL-C (9)	−0.15	0.19	−0.03	−0.18	−0.15	0.86 *	0.59 *	−0.39 *	−	
LDL-C (10)	−0.18	−0.01	0.04	−0.21	−0.02	0.84 *	0.32 *	0.05	0.76 *	−
CHOL/HDL ratio (11)	−0.14	0.03	0.00	−0.23	−0.01	0.71 *	0.36 *	−0.08	0.71 *	0.96 *

Legend: PAL—physical activity level, PL—physical literacy, HL—health literacy, TCHOL—total cholesterol, TG—triglycerides, HDL-C—high-density lipoprotein, non-HDL-C—non-high-density lipoprotein-cholesterol, LDL-C—low-density lipoproteins; * indicates the statistical significance of *p* < 0.05.

**Table 5 medicina-58-01316-t005:** Correlations between studied variables for girls (*n* = 177).

	1	2	3	4	5	6	7	8	9	10
Body height (1)	-									
Body mass (2)	0.54 *	-								
PAL (3)	−0.08	0.07	-							
PL (4)	−0.01	−0.05	0.51 *	-						
HL (5)	−0.01	0.08	0.07	0.31 *	-					
TCHOL (6)	0.01	0.11	0.04	0.06	0.20 *	-				
TG (7)	0.09	0.11	0.01	−0.09	0.11	0.28 *	-			
HDL-C (8)	−0.04	−0.19 *	0.04	0.05	0.06	0.21 *	−0.15	-		
non-HDL-C (9)	0.03	0.21 *	0.02	0.04	−0.25 *	0.88 *	0.35 *	−0.25 *	-	
LDL-C (10)	0.02	0.17 *	0.00	0.01	−0.24 *	0.84 *	0.18 *	−0.09	0.88 *	-
CHOL/HDL ratio (11)	−0.07	0.07	−0.03	−0.04	0.07	0.32 *	0.12	−0.13	0.39 *	0.47 *

Legend: PAL—physical activity level, PL—physical literacy, HL—health literacy, TCHOL—total cholesterol, TG—triglycerides, HDL-C—high-density lipoprotein, non-HDL-C—non-high-density lipoprotein-cholesterol, LDL-C—low-density lipoproteins; * indicates the statistical significance of *p* < 0.05.

## Data Availability

Data will be provided to all interested parties upon reasonable request.

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
