# Peer review of "Specificity of the Associations between Indices of Cardiovascular Health with Health Literacy and Physical Literacy; A Cross-Sectional Study in Older Adolescents"

_medicina, 2022, doi:10.3390/medicina58101316_

Round 1

Reviewer 1 Report

I suggest that the results be presented more clearly. Figure 1 could be presented in a table and not in a bar graph, it is difficult to understand as it is. Tables need to be formatted tables need adjustments, figure 2 is a little hard to read. But after these adjustments I believe the manuscript is ready for publication.

Author Response

Comments and Suggestions for Authors

I suggest that the results be presented more clearly. Figure 1 could be presented in a table and not in a bar graph, it is difficult to understand as it is. Tables need to be formatted tables need adjustments, figure 2 is a little hard to read. But after these adjustments I believe the manuscript is ready for publication.

RESPONSE: Thank you for your support and suggestion. Indeed, original tables and figures were relatively difficult to understand. We believe that we improved it. In brief, as you suggested, data originally presented in Figure 1 are now presented in Table as you suggested (please see Table 2 now). Also, Figures 2 and 3 (now Figure 1 and 2) are improved and hopefully clearer now.

Reviewer 2 Report

Comments:

Line 33- double comma (“,,”) prior to people’s

Line 57- COVID spelled incorrectly (“COVOD”)

Please clarify, if qualifying participants were disqualified if the presented COVID-19 symptoms at the time of testing or if they were ever COVID-19 positive.

Did you use any positive controls to ensure the Mission Cholesterol Test Device was accurate or validate the meter was operating accurately? There is a study published that indicates the Mission Cholesterol meter for serum lipids had analytical biases for triglycerides and total cholesterol [Quartey et. al, Evaluation of the analytical performance of the mission cholesterol meter for serum lipids using NCEP criteria. International J of Medical and Health Research, 2019, 6, 137-139.]

Line 180 are you referring to Table 1 when you state “(please see previously for details)?

Line 320-321 awkward sentence – “This is particular possible in our sample, where we evidenced no association between PAL and lipid profile…”

Was any  of your lipid results confirmed using a laboratory or additional analytical equipment used for blood draws?

Author contributions are missing.

Remove Acknowledgements if there are none.

Author Response

Line 33- double comma (“,,”) prior to people’s

RESPONSE: Double comma is there to indicate the quotation. However, we changed the style of the double comma, so it is more visible.

Line 57- COVID spelled incorrectly (“COVOD”)

RESPONSE: Corrected.

Please clarify, if qualifying participants were disqualified if the presented COVID-19 symptoms at the time of testing or if they were ever COVID-19 positive.

RESPONSE: It is now clarified. Text now reads: “Potential participants with (or suspected of) acute inflammatory disease (e.g., COVID-19) at the time of testing were excluded from the study”.

Did you use any positive controls to ensure the Mission Cholesterol Test Device was accurate or validate the meter was operating accurately? There is a study published that indicates the Mission Cholesterol meter for serum lipids had analytical biases for triglycerides and total cholesterol [Quartey et. al, Evaluation of the analytical performance of the mission cholesterol meter for serum lipids using NCEP criteria. International J of Medical and Health Research, 2019, 6, 137-139.]

RESPONSE: For the purpose of this study we purchased a brand-new Mission Cholesterol test Device (serial number: 293C10015FD) and we assumed the device has passed validation and technical control prior buying it. Before starting sampling, we observed the guidelines of manufacturer's protocol – with each new box of the test device we did calibration following the manuals – we inserted the cod chip that automatically calibrates the meter (please see https://www.swisspointofcare.com/wp-content/uploads/2019/08/Mission-Cholesterol-3in1-Manual-EN-2.pdf . This is now specified in the Variables subsection (please see highlighted text; line 162 onward).

However, we find several other studies about accuracy and diagnostic performance of commercially-available self-tests POC devices including Mission Cholesterol test Device (please see http://dx.doi.org/10.26717/BJSTR.2021.35.005735; https://doi.org/10.1177/0004563221992393). We are aware of reported analytical biases. Also, we agree that self-test devices need better regulation and standardization so this could be the limitation of our study.  Thus, we added this as a limitation of our study. Text reads: “Moreover, another limitation can be the usage of the Mission Cholesterol Test Device, which might have analytical biases for some of the analysed variables.” (please see Section 4.3. Limitations and strengths).

Line 180 are you referring to Table 1 when you state “(please see previously for details)?

RESPONSE: Yes, it is now clarified in the text.

Line 320-321 awkward sentence – “This is particular possible in our sample, where we evidenced no association between PAL and lipid profile…”

RESPONSE: We tried to make the sentence clearer. Text now reads: “No association between PAL and lipid profile in our sample (please see results for more details) can also be explained by numerous factors influencing lipid profile, and not only PAL.”

Was any  of your lipid results confirmed using a laboratory or additional analytical equipment used for blood draws?

RESPONSE: We understand your concern; however, our results were not confirmed by any other laboratory or analytical equipment. The reason for this is that our measurement was done  on-site in school settings and during school time. We estimated that participants would feel better in a familiar setting - school. Also, the study was conducted during the COVID -19 pandemic, so we avoided the possibility of participants being exposed to COVID-19 or any other disease in hospital/ referent laboratory settings.

Further, in planning our research, we decided to avoid the transport of blood samples to the referent laboratory. Indeed, the Brasilian study confirmed that processes and handling of blood samples in the pre-analytical phase, for example, may be responsible for 32% to 75% of the total variation of results in laboratory testing. (please see: https://doi.org/10.1590/0102-311X00122816).

This has also been added as a limitation of our study. Text reads: “Finally, lipid results were not confirmed using laboratory or additional analytical equipment, but this was a consequence of testing protocol which included testing in school settings and during school time. Namely, the study was done during COVID-19 pandemic, so we tried to avoid the possibility of participants being exposed to COVID-19 or any other disease in hospital/ referent laboratory settings. (please see Section 4.3. Limitations and strengths).

Author contributions are missing.

RESPONSE: Thank you, added.

Remove Acknowledgements if there are none.

RESPONSE: We added Acknowledgements. Text reads “Authors are grateful to school authorities and laboratory technicians for their help and support.”

Staying at your disposal!